# Corrosion engineering towards efficient oxygen evolution electrodes with stable catalytic activity for over 6000 hours

Yipu Liu[1], Xiao Liang[1], Lin Gu[2], Yu Zhang[3], Guo-Dong Li[1], Xiaoxin Zou [1] & Jie-Sheng Chen[4]

Although a number of nonprecious materials can exhibit catalytic activity approaching (sometimes even outperforming) that of iridium oxide catalysts for the oxygen evolution reaction, their catalytic lifetimes rarely exceed more than several hundred hours under operating conditions. Here we develop an energy-efficient, cost-effective, scaled-up corrosion engineering method for transforming inexpensive iron substrates (e.g., iron plate and iron foam) into highly active and ultrastable electrodes for oxygen evolution reaction. This synthetic method is achieved via a desired corrosion reaction of iron substrates with oxygen in aqueous solutions containing divalent cations (e.g., nickel) at ambient temperature. This process results in the growth on iron substrates of thin film nanosheet arrays that consist of iron-containing layered double hydroxides, instead of rust. This inexpensive and simple manufacturing technique affords iron-substrate-derived electrodes possessing excellent catalytic activities and activity retention for over 6000 hours at 1000 mA cm$^{-2}$ current densities.

[1] State Key Laboratory of Inorganic Synthesis and Preparative Chemistry, College of Chemistry, Jilin University, 130012 Changchun, China. [2] Institute of Physics, Chinese Academy of Sciences, 100190 Beijing, China. [3] School of Chemistry, Beihang University, 100191 Beijing, China. [4] School of Chemistry and Chemical Engineering, Shanghai Jiao Tong University, 200240 Shanghai, China. Correspondence and requests for materials should be addressed to X.Z. (email: xxzou@jlu.edu.cn)

The electrochemical water splitting has long been considered a promising approach to the production of clean hydrogen fuel by using renewable energy sources[1, 2]. The oxygen evolution reaction (OER) and hydrogen evolution reaction (HER) are the primary half-reactions that occur at the anode and cathode respectively in the electrochemical water splitting reaction. Compared with HER, OER is a more energy-intensive process in the water splitting reaction, due to intrinsically more complex, multiple proton/electron-coupled steps involved in this half-reaction[3–5]. As a consequence, efficient OER electrocatalysis is important for the overall efficiency of the water splitting reaction, and thus oxygen evolution electrodes (or electrocatalysts) with sufficient catalytic activity and stability are urgently demanded. Recently, a number of nonprecious materials have been reported to exhibit catalytic activity approaching (sometimes even outperforming) that of the noble-metal-based IrO$_2$ catalysts for OER[3–5]. The promising nonprecious oxygen evolution electrocatalysts mainly include crystalline/amorphous multimetallic oxides[6–11], layered double hydroxides[12–19], spinel-type oxides[20–23], perovskite-type oxides[24–26], etc.

Although some advances have been obtained in the improvement of catalytic activity for OER and in the mechanistic understanding of activity improvement[3–26], there are still major challenges to employing these promising oxygen evolution electrodes for large-scale, practical applications[27, 28]. The main one is the long-time operational stability: most of recently developed oxygen evolution electrodes are shown to only work well for a few to several tens of hours at small current densities (e.g., 10 mA cm$^{-2}$), and very few can last for several hundreds of hours[6, 8]. The stability problem becomes even more severe when the oxygen evolution electrocatalysts are forced to deliver large catalytic current densities (e.g., 1000 mA cm$^{-2}$: a more practically-relevant value in water splitting devices[11, 28–30].) Their deactivation during OER can be caused by various adverse microstructural evolutions of catalytic active phases, such as oxidative decomposition, structural reconstruction, metal leaching and irregular aggregation. Another non-negligible stability problem is the peeling of catalytic active species from current collector during OER, especially at large catalytic current densities. Therefore, developing highly active oxygen evolution electrodes that can possess significantly prolonged catalytic lifetime (e.g., beyond thousands of hours) still remains a great challenge.

Herein, we report the fabrication of efficient oxygen evolution electrodes that exhibit catalytic activity comparable to those of the state-of-the-art, nonprecious ones, while remaining highly stable for more than 6000 h (>8 months) at a current density of 1000 mA cm$^{-2}$. The electrodes are fabricated from inexpensive iron substrates through a corrosion engineering strategy without any additional energy consumption. To the best of our knowledge, there is no example of efficient and stable OER electrocatalysis in the time span of several months at large current densities. Notably also, an iron corrosion reaction that is usually considered to be negative and unwanted has never been shown to be a positive and desired one for the fabrication of LDHs-based oxygen evolution electrodes.

## Results

**Fabrication of the electrodes via corrosion engineering strategy.** Iron corrosion is a common phenomenon. When iron substrates are brought in contact with water and air, iron rust often unavoidably forms on the surface of iron substrates. Iron corrosion has been usually looked upon as a bad reaction because it causes functional deterioration of the material and big economic losses. Here we show that iron corrosion is not necessarily harmful for the properties of a material, and meticulous corrosion engineering can endow the material with useful functionalities (high catalytic activity and stability for OER herein) that are not easily achievable by other methods.

Particularly, the well-designed iron corrosion is readily realized via immersing iron substrates (e.g., iron plate) in an aqueous solution containing a certain amount of divalent cations (e.g., Ni$^{2+}$, Co$^{2+}$, Mn$^{2+}$, or Mg$^{2+}$) at ambient temperature. If not particularly indicated, the concentration of divalent cations in the corrosive solution is 100 μmol L$^{-1}$ (see experimental details in Experimental section). Due to the intentional introduction of divalent cations in the corrosive environment, Fe-bearing layered double hydroxides (LDHs) are spontaneously generated on iron substrates, instead of the commonly-formed iron rusts (Fig. 1a). Additionally, the as-generated LDHs exist in the form of well-oriented, grain boundary-enriched nanosheet array thin film (Fig. 1b), whose advantageous microstructural features (i.e., nanosheet array architecture and abundant grain boundaries) are believed to be beneficial for electrochemical reactions.

After corrosive treatment of iron plates with various aqueous solutions containing different divalent cations, the final materials are all characterized by various methods. The material, obtained in the corrosive environment containing Ni$^{2+}$ cations, is discussed here as a representative example in detail (Fig. 2). As shown in a photograph of the material (Fig. 2a, inset), there are not yellow or red rust spots visually observed on the surface of the material, indicating the possible formation of a new corrosion product, instead of usual rusts (Supplementary Fig. 1), on iron plate. This speculation is confirmed by powder X-ray diffraction (XRD, Fig. 2a). The XRD result reveals that besides of the XRD peaks of metallic iron, a set of XRD peaks that are characteristic of the structure of LDHs with R-3m symmetry can be identified[31].

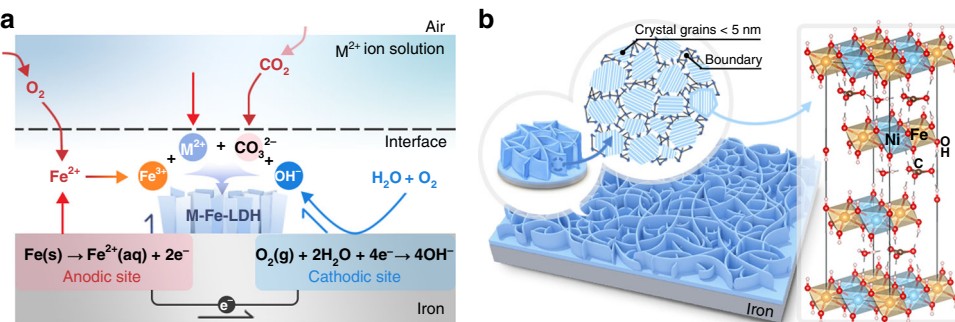

**Fig. 1** Schematic illustrations of the formation and microstructure of the electrodes. **a** The specific reactions happened during the corrosion of iron substrates; and **b** the formation of grain boundary-enriched layered double hydroxide (LDH) nanosheet arrays on the surface of iron substrates. A representative crystal structure of LDH is also shown in **b**

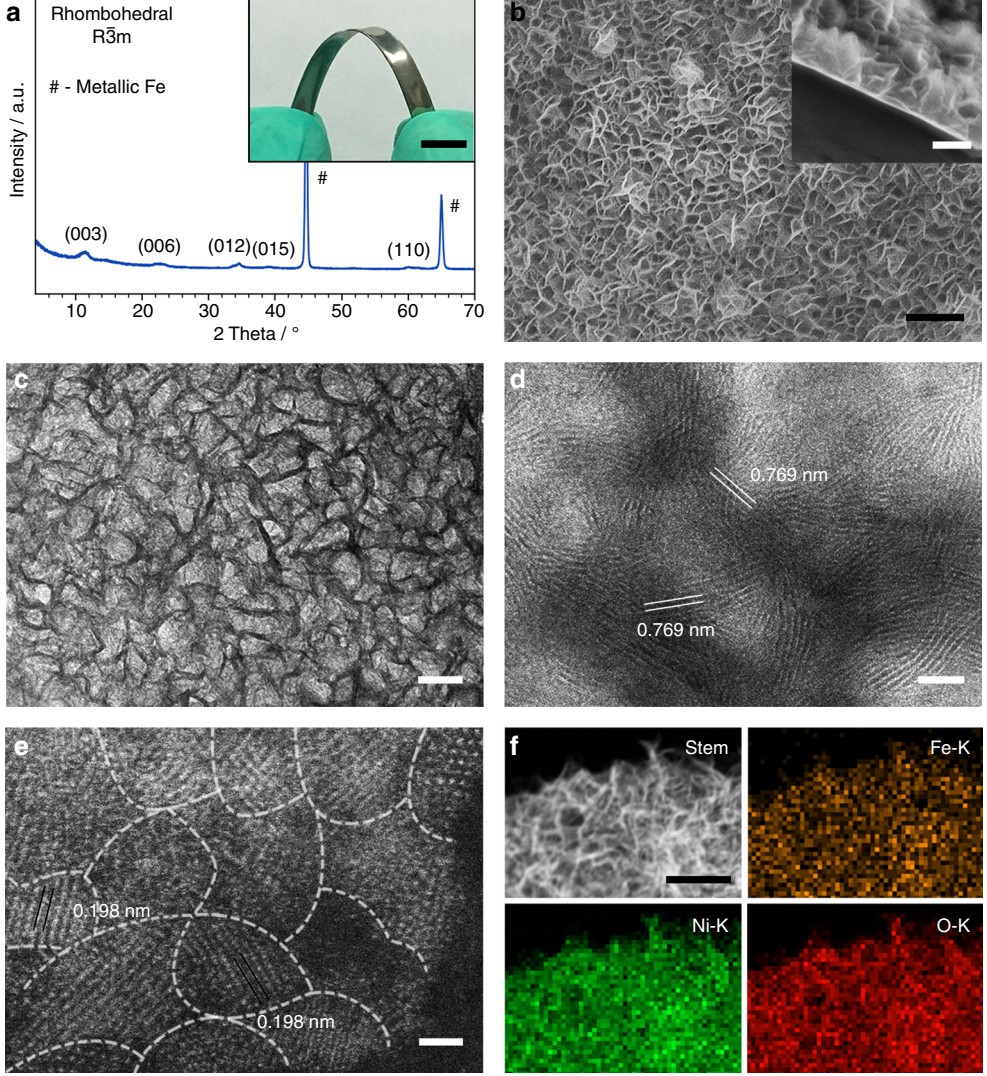

**Fig. 2** Structural Characterizations of the electrodes. **a** XRD pattern of the material obtained by the corrosive treatment of iron plate in the aqueous solution containing $Ni^{2+}$ cations, with its digital image shown in the inset. Scale bar, 1 cm. **b** Top-down and cross-section (inset) SEM images of the material. Scale bars, 400 nm in **b** and 200 nm in the inset. **c** TEM, **d** HRTEM, and **e** HAADF-STEM image of the LDH thin film on the surface of iron plate. **f** STEM image of the LDH thin film and the corresponding elemental mapping images. Scale bars, 50, 5, 1, and 200 nm in **c**, **d**, **e** and **f**, respectively

The interlayer spacing of LDHs is calculated to be *ca.* 0.768 nm from the (003) diffraction peak, and this value is almost the same with those of the LDHs with carbonate as the intercalated anions[15]. This result is in agreement with the recent studies demonstrating the formation of carbonate-intercalated LDHs always preferred when $CO_2$, which would generate carbonate ions in situ, was not intentionally avoided in the reaction system[32, 33]. In addition, the XRD peaks of LDHs for the material appear obviously weaker and broader than those for the powdered LDH particles (Supplementary Fig. 2), suggesting the relatively low crystallinity and small crystalline domains of LDHs in the material.

The LDH thin film on the surface of iron plate is further characterized by the scanning electron microscopy (SEM). As shown in Fig. 2b, the uniform LDH nanosheet array thin film is vertically grown throughout the surface of iron plate with abundant, open voids among nanosheets, and the thickness of the thin film is found to be *ca.* 200 nm. Transmission electron microscopy (TEM) image confirms the nanosheet array structure

again and gives the LDH nanosheet's thickness of *ca.* 8 nm (Fig. 2c). High-resolution TEM (HRTEM) image shows that from the orthographic view of the LDH thin film, a set of lattice fringes with $d = 0.769$ nm that are associated with the interlayer spacing of LDHs can be observed (Fig. 2d). This result suggests that all the LDH nanosheets tend to preferentially grow along the direction parallel to the basal planes of LDHs. High-angle annular dark-field (HAADF) STEM image (Fig. 2e) of an individual LDH nanosheet, coupled with its HRTEM images (Supplementary Fig. 3) reveals that the nanosheet consists of ultrasmall, crystalline nano-domains (<5 nm) as the major constituent and amorphous nano-domains as the minor constituent. This result also suggests that there would be a large proportion of grain boundaries between the crystalline nano-domains. The lattice spacing of the crystalline domains is found to be 0.198 nm, which is associated with the (018) crystallographic planes of LDHs. Moreover, elemental-mapping of the LDH thin film (Fig. 2f) shows that the Fe, Ni and O elements are homogeneously distributed over the thin film. The Fe:Ni atomic ratio in the LDH thin film is

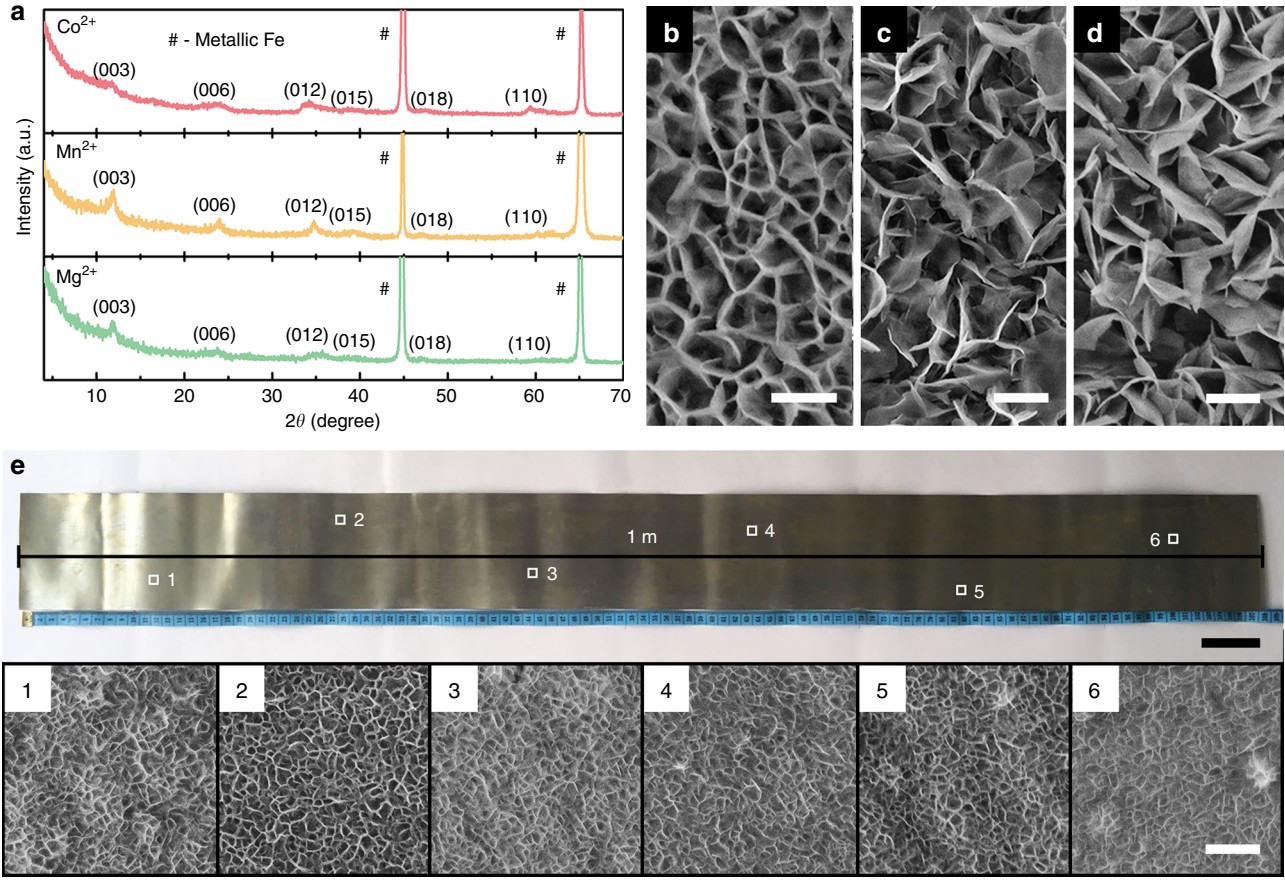

**Fig. 3** Extensibility and scale-up experiment of corrosion engineering method **a** XRD patterns and **b–d** SEM images of the materials obtained by the corrosive treatment of iron plate in the aqueous solution containing $Co^{2+}$, $Mn^{2+}$, or $Mg^{2+}$ cations. Scale bars, 400 nm, 1 μm, and 1 μm in **b**, **c**, **d**, respectively. **e** Digital image of a material (0.1 m × 1 m) obtained from a scaled-up corrosion engineering method in the aqueous solution containing $Ni^{2+}$ cations (Scale bar: 5 cm) and the corresponding SEM images of the representative areas 1–6, respectively. Scale bar, 400 nm

determined to be 1.1:1 by energy dispersive X-ray spectroscopy (EDS), and the Fe and Ni species in the LDH thin film exist in the form of $Fe^{3+}$ and $Ni^{2+}$, as revealed by X-ray photoelectron spectroscopy (XPS, Supplementary Fig. 4)[11].

Besides of $Ni^{2+}$ cations, several other divalent cations (e.g., $Co^{2+}$, $Mn^{2+}$, and $Mg^{2+}$) can also be used to effectively modify the corrosive environment of iron plates. Correspondingly, this modulation leads to the formation of CoFe-LDH, MnFe-LDH and MgFe-LDH in lieu of NiFe-LDH on the surface of iron plates according to the presence of the different divalent cations (Fig. 3a). All the materials also comprise nanosheet array thin films of the corresponding LDHs on iron plates (Fig. 3b–d). These results demonstrate the extensibility of the corrosion engineering method. In addition, this corrosion engineering method is scaled up easily. For instance, a large-area material (0.1 m × 1 m) comprising uniform LDH nanosheet array thin film on iron plate has been prepared (Fig. 3e).

We further attempted to create the corrosive solutions using other cations with lower or higher valence states, such as $Na^+$, $K^+$, $Cr^{3+}$, $Ga^{3+}$, and $Sn^{4+}$, in lieu of divalent cations, but failed to generate LDHs-based nanoarray thin films on iron plates (Supplementary Fig. 5-6). This should be because those cations do not meet the basic structural requirement of LDHs, where the positive layers of LDHs usually consist of both divalent and trivalent cations[34]. Trivalent cations (i.e., $Fe^{3+}$) can be produced in situ from iron plates in the corrosive solution. These results further demonstrate that the presence of divalent cations is

necessary for the formation of LDHs on iron plates, instead of the commonly formed iron rusts.

We also studied the effects of concentration of $Ni^{2+}$ (as a representative divalent cation) on the morphology of the resulting electrodes. In particular, we prepared the corrosive solutions of $Ni^{2+}$ with a wide concentration range from 10 to 200 μmol L$^{-1}$, and correspondingly we fabricated a series of electrode materials by the corrosion engineering method. Structural characterization results (Supplementary Fig. 7-8) reveal that, LDHs-based nanoarray thin films can be generated on iron plates over the whole concentration range from 10 to 200 μmol L$^{-1}$, but a high concentration of $Ni^{2+}$ (typically ≥100 μmol L$^{-1}$) is crucial for the generation of abundant grain boundaries (or low crystallinity) in the LDH nanosheets. The reason behind this phenomenon might lie in the more acidic environment created in the solution containing higher $Ni^{2+}$ concentration (e.g., pH 5.88 for $Ni^{2+}$ concentration of 100 μmol L$^{-1}$, pH 6.28 for $Ni^{2+}$ concentration of 10 μmol L$^{-1}$). In view of the growth preference of LDHs, the more acidic environment should suppress the crystallization of LDHs more, finally leading to the formation of grain boundaries-enriched LDH nanosheets.

The oriented growth of nanosheet array thin films of LDHs on iron plates can be explained by the classic heterogeneous nucleation/growth mechanism[35]. The iron corrosion can produce a large amount of $Fe^{3+}$ and $OH^-$ ions (the precursors for LDHs nucleation) near the surface of iron plates and the heterogeneous nucleation of LDHs is kinetically favorable at the iron-solution

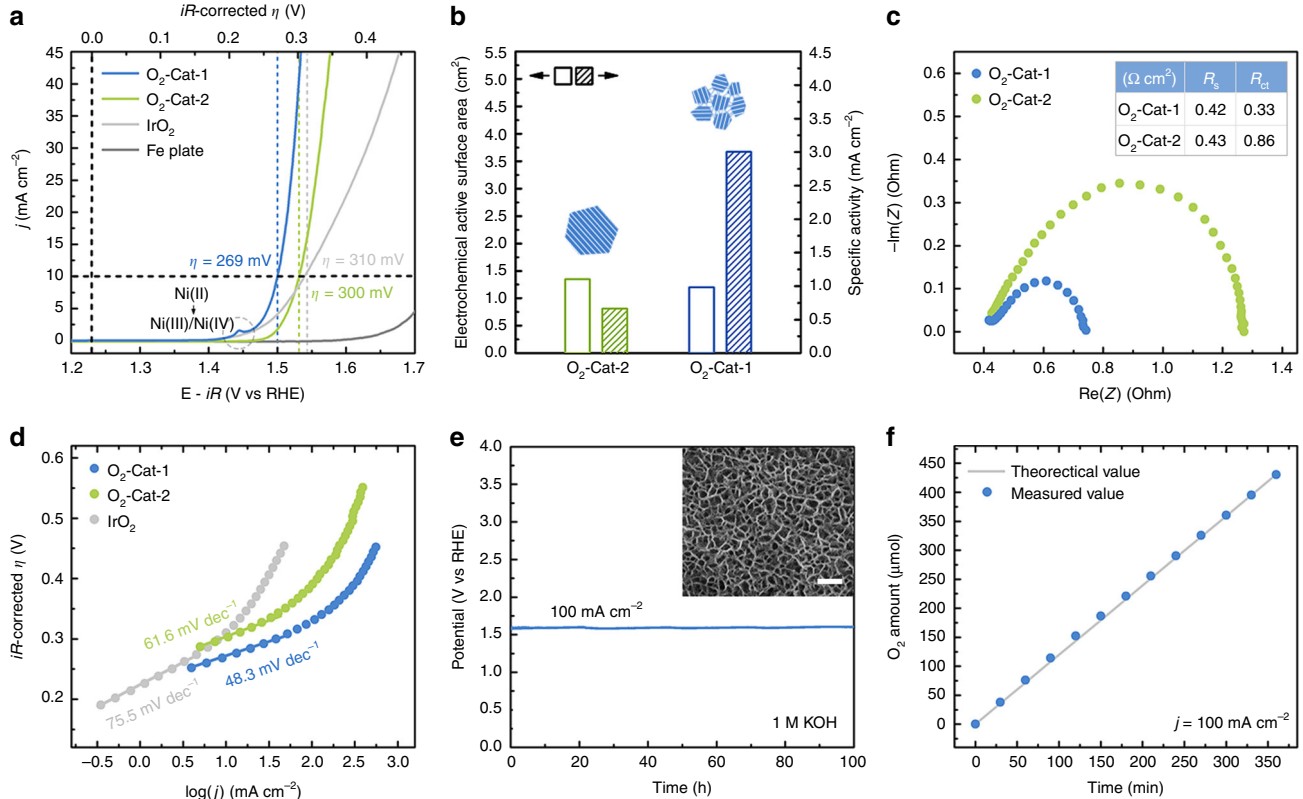

**Fig. 4** Electrocatalytic properties of the electrodes for OER. **a** Polarization curves of $O_2$-Cat-1, $O_2$-Cat-2, $IrO_2$ and iron plate in 1 M KOH electrolyte with 85% *iR*-compensations. The current densities are normalized by the geometric area. **b** Comparison of electrochemical active surface areas (ECSAs) and ECSA-normalized specific activities of $O_2$-Cat-1 and $O_2$-Cat-2 (evaluated at an overpotential of 300 mV). Inset: the schematic illustration for the microstructures of the individual LDH nanosheet in the materials. **c** Comparison of electrochemical impedance spectra between $O_2$-Cat-1 and $O_2$-Cat-2. **d** Tafel plots for OER over $O_2$-Cat-1, $O_2$-Cat-2 and $IrO_2$. **e** Chronopotentiometric curve of $O_2$-Cat-1 in 1 M KOH electrolyte at a current density of 100 mA cm$^{-2}$. Inset: SEM image of $O_2$-Cat-1 after stability test. Scale bar, 200 nm. **f** Electrocatalytic efficiency of oxygen production over $O_2$-Cat-1 at a current density of 100 mA cm$^{-2}$

interface. Due to the anisotropic crystal structure of LDHs, their growth in the *ab*-direction is faster than that in the *c*-direction[35], so that all the LDH nanosheets tend to preferentially grow along the direction parallel to the basal planes of LDHs.

Based on the above experimental data, the divalent cations in the corrosion engineering process should play two important roles in the growth of grain boundary-enriched nanosheet array thin films of LDHs on iron substrates. On the one hand, the divalent cations provide necessary components to meet the basic structural requirement of LDHs, where the positive layers of LDHs usually consist of both divalent and trivalent cations. On the other hand, the divalent cations, especially $Ni^{2+}$, create a suitable, weak acidic solution that is crucial for the formation of abundant grain boundaries in the LDH nanosheets.

Furthermore, we found that the presence of oxygen was required for the formation of LDHs on iron, because the iron plates do not corrode without air (Supplementary Fig. 9). Moreover, further control experiment demonstrates that the presence of $CO_2$ (or $CO_3^{2-}$) was not necessary for the formation of LDHs on the iron (Supplementary Fig. 10), although the carbonate is identified as the charge-balancing anions between the layers of LDHs in the materials we synthesize. By coupling all the above results with previous studies on the rusting of iron[36], a possible reaction mechanism is proposed based on a set of electrochemical reactions that may finally result in the generation of LDHs on iron plates (Fig. 1a). The inherent driving force of

iron corrosion is the electric potential difference between $Fe/Fe^{2+}$ and $OH^-/O_2$, and several electrochemical reactions that should take place simultaneously are described as below:

$$Fe \rightarrow Fe^{2+} + 2e^-$$

$$Fe^{2+} \rightarrow Fe^{3+} + e^-$$

$$O_2 + 2H_2O + 4e^- \rightarrow 4OH^-$$

$$Fe^{3+} + M^{2+} + OH^- + CO_3^{2-} \rightarrow LDH(M = Ni, Co, Mn \text{ or } Mg)$$

It's worth adding here that from the perspective of the preparation of thin films of LDHs, the advantages of the corrosion engineering method can be appreciated by comparing it with the methods developed recently (e.g., electrochemical deposition and hydrothermal synthesis)[11–13, 37]. Those methods must require thermal or electric energy to drive the formation of thin films of LDHs. Those methods must require special reaction equipment and accessories, and they are difficultly scaled up. And those methods cannot use metallic iron as the substrates, because iron rusts are always the main reaction products under these conditions. The above comparisons suggest, the corrosion

engineering method is a simple, yet scalable and cost-effective, way to access this class of materials with multiple functions.

**Electrocatalytic properties of the electrodes for OER**. Next, the electrocatalytic property for OER of the material from the corrosion engineering method (labeled as Route 1) in the aqueous solution containing $Ni^{2+}$ cations are investigated in a detailed way. This is because this particular material, denoted as $O_2$-Cat-1 hereafter, exhibits the highest catalytic activity among the materials we synthesize (Supplementary Fig. 11). For comparative purposes, electrochemical deposition (Route 2) as a prominent method is also applied to produce the corresponding LDH-based electrodes that have been proven to be highly active for OER. The resulting material is correspondingly denoted as $O_2$-Cat-2, and its characterizations are provided in SI (Supplementary Fig. 12-13). It is worth adding here that $O_2$-Cat-2 is composed of LDH nanosheet array thin film on nickel plate (the morphology alike to $O_2$-Cat-1), but their LDH nanosheets with a Ni:Fe atomic ratio of 1.9:1 have relatively larger crystalline domains and less grain boundaries in comparison with those for $O_2$-Cat-1.

Figure 4a shows the polarization curve of $O_2$-Cat-1 for OER. There is an anodic peak at around 1.45 V vs. RHE, which is associated with the redox process of Ni(II)/Ni(III or IV)[14]. In contrast with iron plate's inactivity, $O_2$-Cat-1 exhibits a remarkable electrocatalytic activity for OER, clearly confirming the value-added transformation of inexpensive iron plate into highly efficient electrode for OER. Additionally, $O_2$-Cat-1's electrocatalytic activity even surpasses that of the noble metal-based $IrO_2$ electrocatalyst. More specifically, $O_2$-Cat-1 requires only an overpotential ($\eta$) of 269 mV to achieve a current density of 10 mA $cm^{-2}$, whereas $IrO_2$ needs a higher overpotential (310 mV).

The electrocatalytic activity for OER of $O_2$-Cat-1 is further evaluated by comparing it with $O_2$-Cat-2 (Fig 4a, b). It is worth noting that different from $O_2$-Cat-1, $O_2$-Cat-2 does not exhibit obvious the pre-oxidation peak of Ni. This result is in agreement with that reported in the original study[37]. But different Ni:Fe atomic ratio in two materials (1.1:1 for $O_2$-Cat-1 vs. 1.9:1 for $O_2$-Cat-2) seems not explain why $O_2$-Cat-2 does not exhibit the pre-oxidation peak of Ni. This is because Ni,Fe-based water oxidation catalysts with a wide range of Ni:Fe atomic ratio usually can show the pre-oxidation peak of Ni before the oxygen evolution reaction[7,38]. We speculate that this phenomenon should correlate with the surface microenvironment of Ni species in $O_2$-Cat-2, which ultimately originates from the particular electrochemical synthetic method. But the clear reason (or atomic basis) for this phenomenon is still unclear at current stage.

As shown in Fig. 4a, $O_2$-Cat-1 exhibits a higher geometric area-normalized electrocatalytic activity than $O_2$-Cat-2. For example, at an overpotential of 300 mV, the current densities (normalized by geometric area) of $O_2$-Cat-1 and $O_2$-Cat-2 are 40.2 and 10 mA $cm^{-2}$, respectively. So the electrocatalytic activity of $O_2$-Cat-1 can be considered to be *ca.* four times as high as that of $O_2$-Cat-2. Additionally, their specific electrocatalytic activities are also obtained by normalizing the measured currents with respect to their electrochemical active surface areas (ECSAs). Because of the similar ECSAs of the two materials (Fig. 4b), their specific electrocatalytic activities exhibit almost the same trend with their geometric area-normalized electrocatalytic activities. On the other hand, the similar ECSAs of $O_2$-Cat-1 and $O_2$-Cat-2 indicate their similar densities of catalytic active sites. By combining the higher electrocatalytic activity of $O_2$-Cat-1 compared with $O_2$-Cat-2, it can be concluded that the intrinsic activity of catalytic active sites of $O_2$-Cat-1 is higher than those of $O_2$-Cat-2. The different Ni:Fe atomic ratio in two materials might not account for the better

catalytic activity of $O_2$-Cat-1 because the composition of both materials were in the optimal range for NiFe-based oxygen evolution electrocatalysts[7,39]. The superior intrinsic catalytic activity of $O_2$-Cat-1 should be contributed to its unusual microstructure, owing to the grain boundary-enriched LDH nanosheets comprising ultrasmall crystalline domains. Such grain boundary-enriched LDH nanosheets can expose more edge sites, which generally have a higher degree of unsaturated coordination and thus have been suggested as more active catalytic sites for OER[15,40]. This claim is further supported by the activity comparison of a series of electrodes fabricated by our corrosion engineering method in the corrosive environments containing different $Ni^{2+}$ concentrations (Supplementary Fig. 14). The results show that the electrodes fabricated with high $Ni^{2+}$ concentrations have more abundant grain boundaries (or more defects) and thereby better catalytic activities for OER.

Furthermore, the better electrocatalytic activity of $O_2$-Cat-1 is also reflected by its smaller charge transfer resistance ($R_{ct}$; Fig. 4c) as well as the lower Tafel slope (48 mV $dec^{-1}$) than that (61.6 mV $dec^{-1}$) for $O_2$-Cat-2 (Fig. 4d). A smaller Tafel slope indicates a rapidly boosted current density with the increase of overpotential, and thus is commonly a good sign for electrocatalysts. In case of $O_2$-Cat-2, the oxygen evolution electrocatalysis is considered limiting by the first electron/proton reaction, i.e., adsorption and energy optimization of OH reactants, $(M + OH \rightarrow M - OH + e^-$ together with $M - OH \rightarrow M - OH^*$, where M represents the catalytic active site) based on a Tafel slope around 60 mV $dec^{-1}$[41]. This result also indicates the kinetically sluggish for the association of OH reactants on electrocatalytic active sites in the presence of $O_2$-Cat-2. In case of $O_2$-Cat-1, the oxygen evolution electrocatalysis would be determined by both the first electron/proton reaction and the second electron/proton reaction $(M - OH + OH^- \rightarrow M - O + H_2O + e^-)$, which yields a Tafel slope near 40 mV $dec^{-1}$[42]. By comparing the respective rate-determining reactions during OER between $O_2$-Cat-1 and $O_2$-Cat-2, it's inferred that the grain boundary-enriched $O_2$-Cat-1 efficiently expedites the electrocatalytic kinetics of OER by exposing more unsaturated edge sites that can facilitate the adsorption and activation of reactants [15,40].

Besides of high electrocatalytic activity, $O_2$-Cat-1 also has good electrocatalytic stability for OER. As shown in Fig. 4e, $O_2$-Cat-1 retains its electrocatalytic activity at a current density of 100 mA $cm^{-2}$ for 100 h. After such long-time electrocatalysis, its morphology and microstructure also remain unchanged, as manifested by the SEM (the inset in Fig. 4e) and TEM results (Supplementary Fig. 15). These results also demonstrate that the underlying iron substrates do not further corroded by oxygen during OER. This should be because the underlying iron substrates are covered with stable thin films of LDHs, protecting the iron substrates against further corrosion. Additionally, $O_2$-Cat-1 exhibits a nearly 100% Faradaic efficiency for OER (Fig. 4f), suggesting that the total electron transfer process during electrocatalysis is dominated by the desired OER.

The above results overall demonstrate the meliority of our corrosion engineering method over the electrochemical deposition method and other methods reported recently (Supplementary Table 1) in terms of making highly efficient oxygen evolution electrodes. $O_2$-Cat-1's excellent electrocatalytic performance can be attributed to the following several facts. On the one hand, $O_2$-Cat-1 has the common structural advantages of binder-free nanoarray electrodes and LDHs materials. The well-oriented nanosheet array architecture is beneficial for avoiding the irregular aggregation of catalytic active phase, exposing the catalytic active sites, and facilitating the mass transport during OER (i.e., allowing better supply of reactants to get the active sites and evacuation of gaseous products from the reaction system;

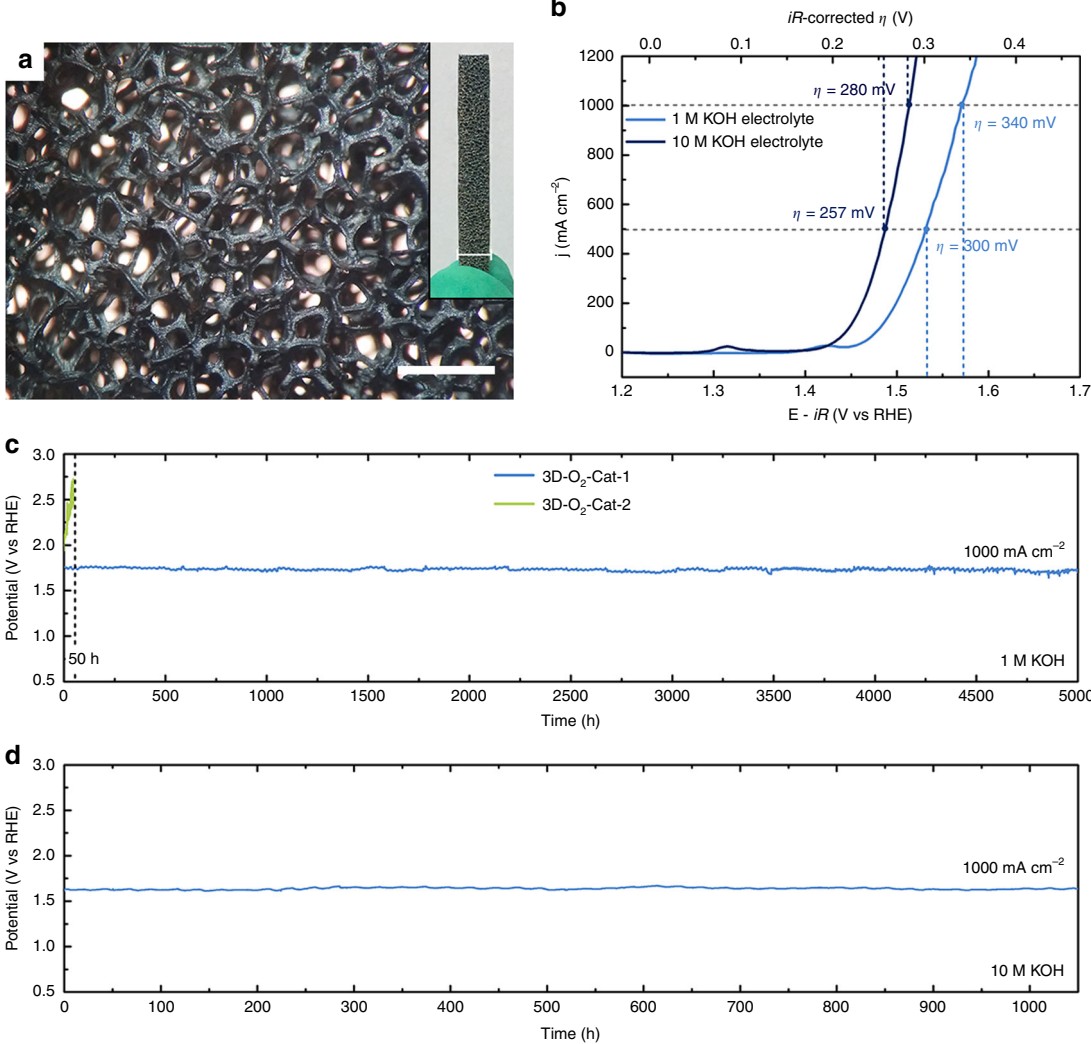

**Fig. 5** Electrocatalytic properties for OER at large current densities. **a** Digital images and **b** *iR*-corrected polarization curves of 3D-O$_2$-Cat-1 in 1 M (light blue line) and 10 M (dark blue line) KOH electrolytes under an extended current window (0-1000 mA cm$^{-2}$). Scale bars, 1 mm and 0.5 cm (inset). **c** Chronopotentiometric curves of 3D-O$_2$-Cat-1 and 3D-O$_2$-Cat-2 in 1 M KOH at a current density of 1000 mA cm$^{-2}$ (without *iR*-correction) **d** Chronopotentiometric curve of 3D-O$_2$-Cat-1 in 10 M KOH at a current density of 1000 mA cm$^{-2}$ (without *iR*-correction)

Supplementary Fig. 16 and Supplementary movie 1). The binder-free structural feature (i.e., binding its catalytically active LDH thin film to the underlying iron substrate without a polymer binder) leads to the presence of an intimate contact between the catalytically active LDH thin film and the current collector (i.e., the underlying iron substrate), further eliminating the interfacial resistance between the two to a large extent. LDHs material usually have good structural stability against oxidative decomposition, structural reconstruction and metal leaching in alkaline media. On the other hand, O$_2$-Cat-1 possesses two particular structural advantages that are not achieved by the electrodes from other synthetic methods (e.g., electrochemical deposition). O$_2$-Cat-1 comprises the LDH nanosheets with ultrasmall crystalline domains and abundant grain boundaries, giving rise to a large proportion of highly active catalytic sites. O$_2$-Cat-1 has the catalytically active LDH thin film that strongly adheres to the iron substrate, avoiding the peeling of catalytically active phases from the current collector during long-term electrocatalysis, even at large current densities (also see below).

As we search for nonprecious oxygen evolution electrodes that might be integrated into practical, large-scale water-splitting technology, they must be able to efficiently deliver large current densities, in the order of 1000 mA cm$^{-2}$ or more. Some researchers even suggest a strict criterion: such electrodes should be capable of achieving a current density of 500 mA cm$^{-2}$ at an overpotential of ≤300 mV[27]. In order to satisfy this requirement, our corrosion engineering method is extended to fabricate a three-dimensional O$_2$-Cat-1 material (denoted as 3D-O$_2$-Cat-1) by replacing iron plate with iron foam (see structural characterization in Fig. 5a and Supplementary Fig. 17). Foam-like substrates with 3D macroporous architecture have been often used recently to increase exposed surface area and loading amount of catalytic active species per geometric area[9, 12]. For comparison, three-dimensional O$_2$-Cat-2 (denoted as 3D-O$_2$-Cat-2) is also prepared according to the previously-reported method (see its digital and SEM images in Supplementary Fig. 18)[37]. As expected, 3D-O$_2$-Cat-1 has a better electrocatalytic activity for OER than 3D-O$_2$-Cat-2 (Supplementary Fig. 19).

As shown in Fig. 5b, 3D-O$_2$-Cat-1 affords current densities of 500 and 1000 mA cm$^{-2}$ at 300 and 340 mV in 1 M KOH solution, respectively. The material requires smaller overpotentials of 257 and 280 mV to achieve the same current densities in 10 M KOH solution, which is employed as the electrolyte in current practical water-alkali electrolyzers[43]. These results demonstrate that the

catalytic activity of 3D-O$_2$-Cat-1 meets the activity requirement in practical water-splitting technology. In addition, to the best of our knowledge, 3D-O$_2$-Cat-1 is one of the most active oxygen evolution electrodes in alkaline media reported to date (Supplementary Table 1) [3–26].

To evaluate the stability of 3D-O$_2$-Cat-1 at large current densities, the electrocatalytic OER is allowed to proceed for 5000 h in 1 M KOH solution (Fig. 5c), and then for 1050 h (Fig. 5d) in 10 M KOH solution at a current density of 1000 mA cm$^{-2}$. For comparison, the catalytic stability of 3D-O$_2$-Cat-2 is also measured in 1 M KOH solution at this current density. The results reveal that although 3D-O$_2$-Cat-2 can retain their catalytic activity at small current densities (Supplementary Fig. 20), this material undergoes serious deactivation over 50 h at large current densities (Fig. 5c). The main cause of the observed serious deactivation is the peeling of catalytic active species (i.e., LDH thin film) from the metal substrates during the intense oxygen evolution process at large current densities (Supplementary Fig. 21). Similarly, the LDHs-based oxygen evolution electrode, fabricated by the hydrothermal deposition method[13], also exhibit an observable deactivation at large current densities (Supplementary Fig. 22). In contrast, 3D-O$_2$-Cat-1 can keep its electrocatalytic activity unchanged for more than 6000 h (>8 months), suggesting the extraordinary structural and catalytic stability of the material for OER.

## Discussion

In summary, a corrosion engineering method has been presented for the value-added transformation of inexpensive iron substrates into high-performance electrodes for OER. The key for the method being successful is the introduction of suitable divalent cations in the corrosive environment, ultimately leading to the generation of nanosheet array thin films of layered double hydroxides, instead of commonly-formed iron rusts, on iron substrates. The iron-substrate-derived electrodes exhibit good catalytic activity toward OER, and show remarkable catalytic stability for more than 6000 h (>8 months) at large current densities. Their excellent catalytic properties can be attributed to their overall conducive structural features, including the nanosheet array architecture, ultrasmall crystalline domains within nanosheets and strongly coupled interface between LDH thin film and iron substrates. Overall, the electrodes presented herein have great potential in large-scale commercial water splitting technology, because they (i) are made from inexpensive starting materials; (ii) are fabricated with easily scalable and energy-efficient synthetic method; and (iii) can work well for long-time OER at large current densities.

## Methods

**Chemicals and reagents**. Fe foam (FF) (thickness: 1.5 mm, bulk density: 0.45 g cm$^{-3}$, number of pores per inch: 90) was purchased from Kunshan Electronic Co., Ltd. Fe plate (FP) (thickness: 0.3 mm) was purchased from Shanghai Baosteel Group Corporation. Ni foam (NF) (thickness: 1.5 mm, bulk density: 0.23 g cm$^{-3}$, number of pores per inch: 110) was purchased from Changsha Lyrun Material Co., Ltd. Ni plate was purchased from Northeast Special Steel Refco Group Ltd. Nickel sulfate hexahydrate (NiSO$_4$·6H$_2$O), potassium hydroxide (KOH) and urea were purchased from Beijing Chemical Factory. Nickel nitrate hexahydrate (Ni(NO$_3$)$_2$·6H$_2$O), Ferric nitrate nonahydrate (Fe(NO$_3$)$_3$·9H$_2$O) and Ferrous sulfate heptahydrate (Fe(SO$_4$)$_2$·7H$_2$O) were purchased from Sinopharm Chemical Reagent Co., Ltd. Nickel chloride hexahydrate (NiCl$_2$·6H$_2$O) was purchased from Tianjin Huadong Chemical Factory. Cobalt sulfate heptahydrate (CoSO$_4$·7H$_2$O) was purchased from Tianjin Guangfu Fine Chemical Research Institute. Manganese sulfate monohydrate (MnSO$_4$·H$_2$O) was purchased from Xilong Chemical Reagent Co., Ltd. Magnesium sulfate (MgSO$_4$) was purchased from Tianjin Tianda Chemical Reagent Factory. Sodium hydroxide (NaOH) was purchased from Sinopharm Chemical Reagent Co., Ltd. Nafion® perfluorinated resin solution was purchased from Sigma-Aldrich. Iridium dioxide (IrO$_2$) was purchased from Shanghai Macklin Biochemical Co., Ltd. Highly purified water (>18 MΩ cm resistivity) was provided by a PALL PURELAB Plus system.

**Corrosion engineering method**. Iron substrates (iron plate or iron foam; 0.5 cm × 6 cm) were washed with acetone and ethanol several times to clean their surfaces for further use. For the preparation of corrosive solution, a certain amount of nickel salts (e.g., NiSO$_4$·6H$_2$O; 100 μmol) was added into 100 mL deionized water to make a transparent Ni$^{2+}$ solution in a conical flask. The pH of the aqueous solution containing NiSO$_4$·6H$_2$O was 5.88. And then the iron substrates were immersed in the corrosive solution at room temperature (~25 °C) for 12 h. When iron plate was used in the corrosion reaction, the resulting material was labeled as O$_2$-Cat-1. When iron foam was used in the corrosion reaction, the resulting material was labeled as 3D-O$_2$-Cat-1. The loadings of LDH species for O$_2$-Cat-1 and 3D-O$_2$-Cat-1 were determined to be about 0.16 and 2.78 mg per geometric surface area. Moreover, a large-area O$_2$-Cat-1 was prepared by immersing a rolled Fe plate (0.1 m × 1 m) into 3000 mL of corrosive solution containing Ni$^{2+}$ (0.1 mol L$^{-1}$) at room temperature for 12 h.

In addition, several other divalent cations (CoSO$_4$·7H$_2$O, MnSO$_4$·H$_2$O or MgSO$_4$; 10 mmol) were also attempted to replace Ni$^{2+}$ cations to generate the corrosive solutions. The pH values of the aqueous solutions containing CoSO$_4$·7H$_2$O, MnSO$_4$·H$_2$O and MgSO$_4$ are 5.90, 5.35, and 6.83, respectively. Correspondingly, the corrosion reactions resulted in the formation of the Co-containing, Mn-containing, or Mg-containing thin films on the iron substrates. For comparative purpose, we attempted to create the corrosive solutions using other cations with lower or higher valence states, such as Na$^+$, K$^+$, Cr$^{3+}$, Ga$^{3+}$, and Sn$^{4+}$, in lieu of divalent cations.

**Fabrication of control electrode materials**. O$_2$-Cat-2 and 3D-O$_2$-Cat-2 were prepared according to the previous report[37]. Briefly, the reaction electrolyte was prepared by dissolving Ni(NO$_3$)$_2$·6H$_2$O (2.18 g, 7.5 mmol) and Fe(SO$_4$)$_2$·7H$_2$O (2.09 g, 7.5 mmol) into 50 mL of deionized water. After mixed under N$_2$ flow, the electrolyte was transferred into an electrochemical cell. For integrating three-electrode configuration, Ni-based substrates (Ni plate or Ni foam) were used as the working electrode. And Pt wire and saturated calomel electrode (SCE) were used as the counter and reference electrode, respectively. After a potentiostatic deposition at a potential of −1.0 V vs. SCE for 300 s, O$_2$-Cat-2 and 3D-O$_2$-Cat-2 were fabricated, depending on whether the use of Ni plate or Ni foam as the substrates.

The LDH-based electrode was synthesized from hydrothermal method according to previous report[13]. Ni(NO$_3$)$_2$·6H$_2$O (0.3 g, 1 mmol), Fe(NO$_3$)$_3$·9H$_2$O (0.4 g, 1 mmol) and urea (0.3 g, 5 mmol) were dissolved into deionized water (80 mL) under vigorous stirring. Then a piece of cleaned Ni foam (0.5 cm × 5 cm) and the as-prepared solution was transferred into an autoclave and maintained at 120 °C for 12 h. The resulting material was washed with ethanol three times and dried in vacuum at room temperature, giving the LDH-based electrode.

**Characterizations**. The powder X-ray diffraction (XRD) patterns were recorded on a Rigaku D/Max 2550 X-ray diffractometer with Cu Kα radiation ($\lambda = 1.5418$ Å). The scanning electron microscope (SEM) images were obtained with a JEOL JSM 6700 F electron microscope. The transmission electron microscope (TEM) images were obtained with a Philips-FEI Tecnai G2S-Twin microscope equipped with a field emission gun operating at 200 kV. High-resolution STEM measurements were performed on an atomic resolution analytical microscope (JEM-ARM 200 F) operating at 200 kV. The X-ray photoelectron spectroscopy (XPS) was performed on an ESCALAB 250 X-ray photoelectron spectrometer with a monochromatic X-ray source (Al Kα $h\nu =$ 1486.6 eV). The energy scale of the spectrometer was calibrated using Au 4f$_{7/2}$, Cu 2p$_{3/2}$ and Ag 3d$_{5/2}$ peak positions. The standard deviation for the binding energy (BE) values was 0.1 eV. Inductively coupled plasma atomic emission spectroscopy (ICP-OES) was performed on a Perkin-Elmer Optima 3300DV ICP spectrometer.

**Electrochemical measurements**. All electrochemical measurements were conducted in a standard three-electrode configuration with a CH Instrument (Model 660E). The materials (O$_2$-Cat-1, O$_2$-Cat-2, etc.) were used as working electrode. Hg/HgO electrode and Pt plate were used as reference and counter electrodes, respectively. Before the electrochemical measurements, a surface area of 0.09 cm$^2$ on each electrode keeps exposed, with the rest of the electrode sealed with hot melt adhesives. In addition, the catalytic activity of powdered IrO$_2$ was measured by loading it on the iron plate with optimal loading amount (0.4 mg for 0.09 cm$^2$, Supplementary Fig. 23), and a polymer binder (Nafion) was also introduced for a purpose of fixation. Although IrO$_2$ is widely used as the benchmark electrocatalyst for OER, this material is not catalytically stable enough in alkaline media. Thus, the long-time stability of IrO$_2$ for OER was not measured here. During the electrochemical measurements, the electrolyte in electrochemical cell was continuously bubbled with N$_2$, and current densities were normalized with the geometric surface areas of the electrodes. For linear sweep voltammetry (LSV) measurements, the scan rate was set to 1 mV s$^{-1}$, and the resistances of the test system were estimated from corresponding single-point impedance measurements and compensated by 85% iR-drop.

The Hg/HgO electrode was calibrated by the method reported by Boettcher and co-workers[44]. Briefly, two Pt electrodes were first polished and cycled in 0.5 M H$_2$SO$_4$ (about ±2 V for 2 h) for cleaning purpose, and then employed as working electrode (WE) and counter electrode (CE) in 10 M KOH electrolyte. The

electrolyte was saturated by hydrogen before use, and continuous $H_2$ was bubbled over the WE during the calibration. To perform the calibration, a series of controlled-potential chronoamperometric curves were carried out around the possible zero current potential (the interconversion between the hydrogen oxidation and hydrogen evolution reaction) determined by a wide-ranged LSV measurement swept in a cathode direction. In such chronoamperometric measurements, each potential is held constant for 300 s to reach a steady-state value, which is a more reliable value avoiding the possible polarization effects and the contribution of capacitive current. As shown in Supplementary Fig. 24, the result shows that the potential of zero net current can be estimated at $-0.983$ V vs. the Hg/HgO electrode, and the relation between the Hg/HgO reference and RHE in 10 M KOH solution can thus be established using the equation: $E_{RHE} = E_{Hg/HgO} + 0.983$ V. By using the same method, the relation between the Hg/HgO reference and RHE in 1 M KOH solution was also established using the equation: $E_{RHE} = E_{Hg/HgO} + 0.926$ V

Electrochemical impedance spectroscopy (EIS) was performed on the materials under the operating conditions for OER. The initial potential at the electrode was set as 1.582 V vs. RHE. A sinusoidal voltage with an amplitude of 5 mV and a scanning frequency ranging from 10,000 to 0.01 Hz were applied to carry out the measurements.

In order to get the effective electrochemical active surface area (ECSA) of a material, a series of cyclic voltammetry (CV) measurements were performed first at various scan rates (20, 40, 60 mV s$^{-1}$, etc.) in the potential window between 1.112 and 1.212 V vs. RHE. The sweep segments of the measurements were set to 10 to ensure consistency. By plotting the difference of current density ($J$) between the anodic and cathodic sweeps ($J_{anodic} - J_{cathodic}$) at 1.162 V vs. RHE against the scan rate, a linear trend was constructed. Then, the geometric double layer capacitance ($C_{dl}$) was easily calculated because $C_{dl}$ is one half the slope value of the fitting line. Finally, the ECSA of catalyst is estimated from the double-layer capacitance according to the equation:

$$ECSA = \frac{C_{dl}}{C_s} \times ASA,$$

where $C_s$ is the specific capacitance of the sample, and ASA is the actual surface area of the electrode. In this work, the value of $C_s$ is estimated to be 0.04 mF cm$^{-2}$.

In order to determine the Faradic efficiency of an electrode in OER, the $O_2$ gas generated by the electrochemical reaction was collected by a water drainage method and its amount (in mol) was calculated using the ideal gas law. The theoretical value was calculated by assuming that 100% of the current output during the reaction was originated from the OER. Faradic efficiency was then obtained by calculating the ratio of the amount of $O_2$ evolved during OER to the amount of $O_2$ expected to generate based on theoretical considerations.

In order to determine whether Fe and Ni ions were leached in the electrolyte at high current density, we used ICP-OES to detect the Fe and Ni ions in the electrolyte during the long-time electrolysis process at 1000 mA cm$^{-2}$ in the presence of 3D-$O_2$-Cat-1. The electrolyte was collected after 2, 4, 6, 8, 10, 34, and 58 h during operation, and the results shown in Supplementary Table 2 reveal that no detectable leached metallic species are present in the electrolyte.

**Data availability**. The authors declare that the data supporting the findings of this study are available within the paper and its supplementary information files.

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

## Acknowledgements

X.Z. acknowledges the financial support from the NSFC 21771079, National Key R&D Program of China, Grant No. 2017YFA0207800, Jilin Province Science and Technology Development Plan 20170101141JC, Young Elite Scientist Sponsorship Program by CAST, Program for JLU Science and Technology Innovative Research Team (JLUSTIRT) and Fok Ying Tung Education Foundation, Grant No.161011.

## Author contributions

X.Z. conceived the idea, analyzed the data and wrote the manuscript. Y.L. synthesized and characterized the materials, analyzed the data and wrote the manuscript. X.L., G.-D. L., Y.Z. and J.-S.C. assisted Y.L. with the materials synthesis, electrochemical measurements and analysis. L.G. performed the TEM measurement. All of the authors have read the manuscript and agree with its content.

## Additional information

**Competing interests:** The authors declare no competing interests.

