## [Peer Review File · Nature Communications]

Reviewers' comments:

Reviewer #1 (Remarks to the Author):

The research work reported a "corrosion engineering" approach for the fabrication of transitional metal layered double hydroxides (LDHs) ultrastable electrodes for OER. The paper is generally well written and quality of the data is good. However, the overall novelty of the work is not sufficient for publication in Nature Commun. From material perspective, NiFe LDHs is a well-known material for OER and is quite efficient and stable. There is nothing unexpected from this work. From fabrication method point of view, the "corrosion engineering" is also well established; similar approach has already been reported to fabricate NiFe LDH from Ni and stainless steel substrates. The performance of the obtained NiFe LDH is as good as expected but not particularly exciting to be published in a Nature Journal.

Some other comments for authors to consider:

1. "zero-energy-consumption fabrication" in the title is vague and misleading. How come a fabrication process consumes zero energy? This is more a presentational gimmick and should be revised.
2. The use of divalent cations and their roles during the corrosion reaction need to be explained. Can cations with lower or higher valence states achieve the same effect, other than divalent cations? How does the concentration of the divalent cations influence the morphology and catalytic activity of the prepared electrode?
3. The reaction in page 10, line 192 is incorrect, the reaction requires 4e.
4. As shown in Fig. 4a and Fig. S12, only O2-Cat-1 samples show the pre-oxidation peak of Ni, but not the O2-Cat-2 samples? In Fig. 4d, the Tafel slopes for O2-Cat-1 and O2-Cat-2 are almost parallel at low current density, suggesting a similar Tafel slope and reaction kinetics, rather what the authors claimed in page 14 (line 264-278)
5. Fig 5b, What is the pH of 10 M KOH, and how the measured potential is calibrated versus RHE in 10 M KOH?

Reviewer #2 (Remarks to the Author):

The present work reported a Zero-Energy-Consumption fabricated NiFe electrode, prepared via "corrosion engineering" method. As the catalyst for water oxidation, the authors claimed the Fe-bearing 3D-O2-Cat-1 is one of most promising oxygen evolution catalysts affords current densities of 500 and 1000 mA/cm² at 300 and 340 mV in 1 M KOH solution, respectively. However, there remain lots of issues necessary to be addressed. Therefore, I would like to recommend a major revision on this manuscript before acceptance. Please find below are the detail comments:

- 1, What is the role of the divalent cations (e.g., Ni²⁺, Co²⁺, Mn²⁺ or Mg²⁺) in the 'corrosion engineering' process? The author should present a more convincing explanation to clarify them.
- 2, What is the role of the CO₂ and/or CO₃²⁻ in the fabrication process of catalyst?
- 3, For the preparation of the corrosive solution, a certain amount of divalent cations (NiSO₄·6H₂O, Ni(NO₃)₂·6H₂O, NiCl₂·6H₂O, CoSO₄·7H₂O, MnSO₄·H₂O or MgSO₄; 10 mmol) was added into 100 mL deionized water to generate the corrosive solutions. The authors declare it is 'a neutral aqueous solution'. Please check, without neutralization most of the solutions should be acidic, except the one of MgSO₄.
- 4, NiFe based material is a common material in the system under consideration, the authors should focus on defects and morphology introduced here. What is the actual formation mechanism

of such a Fe-bearing structure ?

5, The atomic ratio plays a key role in determining the catalytic performance of NiFe based OER catalyst. The author should provide more information of Ni:Fe atomic ration of O2-Cat-2.

6, Fig. 4a shows the polarization curves of O2-Cat-1 and O2-Cat-2 for OER. If O2-Cat-1 and O2-Cat-2 show similar LDH structure, contain the same Ni:Fe atomic ratio, the intensity of the redox peaks of Ni(II)/Ni(III or IV) should be similar to each other. Why they are totally different?

7, In the introduction, the authors conclude that the deactivation of an OER catalyst during OER can be caused by various adverse microstructural evolutions of catalytic active phases, such as oxidative decomposition, structural reconstruction, metal leaching, irregular aggregation and the peeling of catalytic active species from the current collector, especially at large catalytic current densities. The author should analyze the stability of the catalyst from the five factors during the long-time electrolysis process.

Reviewer #3 (Remarks to the Author):

In this work, Zou et al report the spontaneous corrosion reaction of iron to fabricate efficient oxygen evolution electrode without needing any additional energy inputs (heat or electricity). The resulting electrode is among the most active electrodes for OER in alkaline electrolyte reported to date. Moreover, the resulting electrode exhibits a record long-term catalytic stability for more than 6000 hours, even at a large current density of 1000 mA/cm². Both the synthetic method and the record long-term catalytic stability in this work are impressive. Thus, I recommend the publication of this paper in Nature Communications after some revisions.

1. How the authors got rid of the bubbles effect. At high current densities, the formation of bubbles usually generates very noisy LSV data but the curves shown in this paper are very smooth and precise.

2. It should be better to give the mass loadings of catalysts (i.e., LDH species).

3. As described in the paper, when iron substrates are brought in contact with water and air, iron is ready to be corroded. Could the authors give some explanation why the electrodes (Cat-O2-1 and 3D-Cat-O2-1) could be so stable during OER? In other words, could the underlying iron substrates in these electrodes be further corroded by oxygen during OER?

4. Please provide the characterization results (e.g., SEM images) of 3D-Cat-O2-2.

5. In addition, IrO₂ is widely considered as the benchmarking OER electrocatalyst because of its high catalytic activity in acidic and alkaline media. However, this catalyst is not stable enough in alkaline electrolyte. This is one of the reasons why IrO₂ is not used in alkaline electrolyzer. The authors should point out this drawback of IrO₂ in the paper to show more complete information for the readers.

吉林大学 化学学院

Department of Chemistry

Jilin University

Changchun 130012, China

Dr. Prof. Xiaoxin Zou

State Key Lab. Inorg. Synth. & Prep. Chem.

Tel: +86-431-85168221

xxzou@jlu.edu.cn

<http://zouxxgroup.com/>

Reviewer's Comments and Our Responses:

Reviewer: 1

Comment 1: The research work reported a “corrosion engineering” approach for the fabrication of transitional metal layered double hydroxides (LDHs) ultrastable electrodes for OER. The paper is generally well written and quality of the data is good. However, the overall novelty of the work is not sufficient for publication in Nature Commun. From material perspective, NiFe LDHs is a well-known material for OER and is quite efficient and stable. There is nothing unexpected from this work. From fabrication method point of view, the “corrosion engineering” is also well established; similar approach has already been reported to fabricate NiFe LDH from Ni and stainless steel substrates. The performance of the obtained NiFe LDH is as good as expected but not particularly exciting to be published in a Nature Journal.

Response 1: We thank this reviewer for his/her comments here! We particularly appreciate his/her positive comment: the paper is generally well written and quality of the data is good.

Our response to the reviewer’s concern on the material performance: We would like to emphasize that the primary motivation of this work is to demonstrate a stable oxygen evolution electrode that can possess catalytic lifetime beyond thousands of hours (or several months). We would like to claim again that there is no example of efficient and stable OER electrocatalysis in the time span of several months at large current densities, regardless of the electrodes/electrocatalysts used, including NiFe LDHs and other materials.

I agree with the reviewer’s point that NiFe LDHs are among the most active and stable materials for OER. However, many of recently developed oxygen evolution electrodes based on powdered NiFe LDHs often underwent detectable activation loss over a few to several tens of hours at small current densities (*e.g.*, 10 mA/cm²) (see some examples: *e.g.*, *Nat. Commun.* 2014, 5, 4477, *Adv. Mater.* 2015, 27, 4516-4522, *Adv. Mater.* 2017, 29, 1700017, *Angew. Chem. Int. Ed.* 2018, 57, 172-176). On the other hand, although several binder-free NiFe LDHs thin films have been shown better catalytic activity and stability than the electrodes based on powdered NiFe LDHs, they cannot deliver large current densities stably (*e.g.*, 1000 mA/cm²: a more practically-relevant value in water splitting devices). This claim has been supported by our control experiments (see Figures 5c and S21). Two NiFe LDHs thin film electrodes were fabricated by the two most prominent methods, electrochemical deposition (Figure 5c) and hydrothermal deposition (Figure S21), and they both exhibited poor catalytic stability at large current densities.

吉林大学 化学学院

Department of Chemistry

Jilin University

Changchun 130012, China

Dr. Prof. Xiaoxin Zou

State Key Lab. Inorg. Synth. & Prep. Chem.

Tel: +86-431-85168221

xxzou@jlu.edu.cn

<http://zouxxgroup.com/>

Our response to the reviewer's concern on the novelty of fabrication method: After careful literature searching, we did not find any reports demonstrating the "corrosion engineering" method for the fabrication of NiFe LDHs thin film from Ni and stainless steel substrates. We have to reiterate that this is the first time herein that an iron corrosion reaction that is usually considered to be negative and unwanted can be a positive and desired one for the fabrication of LDHs-based oxygen evolution electrodes.

Although we found that hydrothermal treatment of stainless steel and chemical corrosion of Ni with some chemicals such as nitrate (see literatures: *e.g.*, *Angew. Chem. Int. Ed.* 2016, 55, 9937-9941, *Nano Res.* 2017, 10.1007/s12274-017-1886-7) were used for the fabrication of oxygen evolution electrodes, in those cases amorphous metal (hydro)oxide layers, rather than NiFe LDHs nanoarray thin films, on the metal substrates were generated. Those electrodes reported previously were shown to be active for OER, but their catalytic activities are lower than that of our material. Additionally, these electrodes were never reported to electrocatalyze the OER for thousands of hours. Finally, it is worth noting that an iron corrosion reaction with oxygen and water at room temperature (*i.e.*, totally different corrosion mechanism from the former two cases) has never been employed for the fabrication of high-performance oxygen evolution reaction before. Our work also demonstrates the value-added transformation of inexpensive iron plate into highly efficient electrode for OER.

Overall, we believe that our work will be of considerable interest to the wide readership of *Nature Communications*.

Comment 2: "zero-energy-consumption fabrication" in the title is vague and misleading. How come a fabrication process consumes zero energy? This is more a presentational gimmick and should be revised.

Response 2: The title has been modified to "'Corrosion Engineering' Strategy for Fabricating Efficient Oxygen Evolution Electrodes with Stable Catalytic Activity for > 6,000 Hours".

Comment 3: The use of divalent cations and their roles during the corrosion reaction need to be explained. Can cations with lower or higher valence states achieve the same effect, other than divalent cations? How does the concentration of the divalent cations influence the morphology and catalytic activity of the prepared electrode?

Response 3: We thank for the reviewer's valuable questions.

Our response to the reviewer's question 1: We further attempted to create the corrosive solutions using other cations with lower or higher valence states, such as Na^+ , K^+ , Cr^{3+} , Ga^{3+} and Sn^{4+} , in lieu of divalent cations, but failed to generate LDHs-based nanoarray thin films on iron plates (see Figure S5-S6 in SI). This should be because those cations do not meet the basic structural requirement of LDHs, where the positive layers of LDHs usually consist of both

吉林大学 化学学院

Department of Chemistry

Jilin University

Changchun 130012, China

Dr. Prof. Xiaoxin Zou

State Key Lab. Inorg. Synth. & Prep. Chem.

Tel: +86-431-85168221

xxzou@jlu.edu.cn

http://zouxxgroup.com/

divalent and trivalent cations. Trivalent cations (*i.e.*, Fe³⁺) can be generated *in situ* from iron plates in the corrosive solution. These results further demonstrate that the presence of divalent cations is necessary for the formation of LDHs on iron plates.

Our response to the reviewer's question 2: We studied the effects of concentration of Ni²⁺ (as a representative divalent cation) on the morphology and catalytic activity of the resulting electrodes. In particular, we prepared the corrosive solutions of Ni²⁺ with a wide concentration range from 10 to 200 μmol/L, and correspondingly we fabricated a series of electrode materials by the method presented in the paper. Structural characterization results (Figure S7-8 in SI) reveal that, LDHs-based nanoarray thin films can be generated on iron plates over the whole concentration range from 10 to 200 μmol/L, but a high concentration of Ni²⁺ (typically ≥ 100 μmol/L) is crucial for the generation of rich grain boundaries (or low crystallinity) in the LDH nanosheets. Additionally, catalytic results (Figure S13 in SI) reveal that the electrodes fabricated in the solution containing a high concentration of Ni²⁺ (typically > 100 μmol/L) exhibit better catalytic activity for OER than those fabricated in the solution containing a low concentration of Ni²⁺ thanks to the abundant grain boundaries in the former. Note that the electrode fabricated in the solution containing a Ni²⁺ concentration of 100 μmol/L is selected as the representative material and discussed in a detailed way in our paper.

Comment 4: The reaction in page 10, line 192 is incorrect, the reaction requires 4e.

Response 4: Thanks. We have corrected this mistake in the revised manuscript.

Comment 5: As shown in Fig. 4a and Fig. S12, only O₂-Cat-1 samples show the pre-oxidation peak of Ni, but not the O₂-Cat-2 samples? In Fig. 4d, the Tafel slopes for O₂-Cat-1 and O₂-Cat-2 are almost parallel at low current density, suggesting a similar Tafel slope and reaction kinetics, rather what the authors claimed in page 14 (line 264-278)

Response 5: Thanks for the reviewer's questions.

Our response to the question 1: O₂-Cat-2 does not exhibit obvious the pre-oxidation peak of Ni. Our result is in agreement with that reported in the original study (*Chem. Sci.* 2015, 6, 6624-6631). We speculate that this phenomenon should correlate with the surface microenvironment of Ni species in the material, which ultimately originates from the particular electrochemical synthetic method. But the clear reasons (or atomic basis) for this phenomenon are still unclear at current stage.

Our response to the question 2: Although the Tafel plots for O₂-Cat-1 and O₂-Cat-2 look similar in the figure, but their Tafel slopes are obviously different in fact. The Tafel slopes for O₂-Cat-1 and O₂-Cat-2 are 48.3 and 61.6 mV/dec, respectively (those results are repeatable). Our discussion on the Tafel slopes are based on the previous OER mechanism studies. (*J. Am. Chem. Soc.* 2015, 137, 3638-3648, *ACS Nano* 2017, 11, 5500-5509) The key steps during OER and the

吉林大学 化学学院

Department of Chemistry

Jilin University

Changchun 130012, China

Dr. Prof. Xiaoxin Zou

State Key Lab. Inorg. Synth. & Prep. Chem.

Tel: +86-431-85168221

xxzou@jlu.edu.cn

http://zouxxgroup.com/

related Tafel slopes are shown as below. A modified discussion has been provided in the revised manuscript (or see below).

“...A smaller Tafel slope indicates a rapidly boosted current density with the increase of overpotential, and thus is commonly a good sign for electrocatalysts. In case of O₂-Cat-2, the oxygen evolution electrocatalysis is considered limiting by the first electron/proton reaction, *i.e.*, adsorption and energy optimization of OH reactants, ($M + OH^- \rightarrow M-OH + e^-$ together with $M-OH \rightarrow M-OH^*$, where M represents the catalytic active site) based on a Tafel slope around 60 mV/dec.⁴⁰ This result also indicates the kinetically sluggish for the association of OH reactants on electrocatalytic active sites in the presence of O₂-Cat-2. In case of O₂-Cat-1, the oxygen evolution electrocatalysis would be determined by both the first electron/proton reaction and the second electron/proton reaction ($M-OH + OH^- \rightarrow M-O + H_2O + e^-$), which yields a Tafel slope near 40 mV/dec.⁴¹ By comparing the respective rate-determining reactions during OER between O₂-Cat-1 and O₂-Cat-2, it's inferred that the grain boundary-enriched O₂-Cat-1 efficiently expedites the electrocatalytic kinetics of OER by exposing more unsaturated edge sites that can facilitate the adsorption and activation of reactants.^{14,39} ...”

Comment 6: Fig 5b, What is the pH of 10 M KOH, and how the measured potential is calibrated versus RHE in 10 M KOH?

Response 6: Thanks for these good questions!

Our response to the question 1: The pH value of solution is defined as the decimal logarithm of the reciprocal of the hydrogen ion activity (a_{H^+}) in the solution. Generally, it is easy to measure the pH value of KOH solution with a concentration below 1 mol/L (or 1M) by commercially available pH meter. However, the pH value of 10 M KOH solution is outside of the normal 0-14 range. It is unachievable to accurately measure its pH value at present by the state-of-the-art pH meters. Thus, we cannot provide the accurate pH value of 10 M KOH in the paper. As an alternative approach, we directly calibrate our reference electrode *versus* a reversible hydrogen electrode (RHE) in 10 KOH solution for the better presentation of the electrochemical results.

吉林大学 化学学院

Department of Chemistry

Jilin University

Changchun 130012, China

Dr. Prof. Xiaoxin Zou

State Key Lab. Inorg. Synth. & Prep. Chem.

Tel: +86-431-85168221

xxzou@jlu.edu.cn

<http://zouxxgroup.com/>

Our response to the question 2: The relation between the Hg/HgO reference and RHE can be formulated by $E_{\text{RHE}} = E_{\text{Hg/HgO}} + 0.983 \text{ V}$ in 10 M KOH solution. The detailed procedure of the RHE calibration is provided in the Supporting Information or see below.

“To convert the measured potential *versus* the Hg/HgO electrode into the potential *versus* reversible hydrogen electrode (RHE), the Hg/HgO electrode as reference electrode was calibrated using RHE in 10 M KOH solution. The RHE was constructed according to the previous work reported by Boettcher and co-workers (*Chem. Mater.* 2017, 29, 120-140). Briefly, two Pt electrodes were first polished and cycled in 0.5 M H₂SO₄ (about $\pm 2 \text{ V}$ for 2 hours) for cleaning purpose, and then employed as working electrode (WE) and counter electrode (CE) in 10 M KOH electrolyte. The electrolyte was saturated by hydrogen before use, and continuous H₂ was bubbled over the WE during the calibration. To perform the calibration, a series of controlled-potential chronoamperometric curves were carried out around the possible zero current potential (the interconversion between the hydrogen oxidation and hydrogen evolution reaction) determined by a wide-ranged LSV measurement swept in a cathode direction. In such chronoamperometric measurements, each potential is held constant for 300 s to reach a steady-state value, which is a more reliable value avoiding the possible polarization effects and the contribution of capacitive current. As shown in Figure S23, the result shows that the potential of zero net current can be estimated at -0.983 V *versus* the Hg/HgO electrode, and the relation between the Hg/HgO reference and RHE in 10 M KOH solution can thus be established using eq 2:

$$E_{\text{RHE}} = E_{\text{Hg/HgO}} + 0.983 \text{ V} \quad (2)$$

Reviewer: 2

Comment 1: The present work reported a zero-energy-consumption fabricated NiFe electrode, prepared via “corrosion engineering” method. As the catalyst for water oxidation, the authors claimed the Fe-bearing 3D-O₂-Cat-1 is one of most promising oxygen evolution catalysts affords current densities of 500 and 1000 mA/cm² at 300 and 340 mV in 1 M KOH solution, respectively. However, there remain lots of issues necessary to be addressed. Therefore, I would like to recommend a major revision on this manuscript before acceptance. Please find below are the detail comments:

吉林大学 化学学院

Department of Chemistry

Jilin University

Changchun 130012, China

Dr. Prof. Xiaoxin Zou

State Key Lab. Inorg. Synth. & Prep. Chem.

Tel: +86-431-85168221

xxzou@jlu.edu.cn

<http://zouxxgroup.com/>

Response 1: We thank the reviewer's comments here! We have addressed the comments point-by-point as follows.

Comment 2: What is the role of the divalent cations (e.g., Ni^{2+} , Co^{2+} , Mn^{2+} or Mg^{2+}) in the 'corrosion engineering' process? The author should present a more convincing explanation to clarify them.

Response 2: The divalent cations in the "corrosion engineering" process play two important roles in the growth of grain boundary-enriched nanosheet array thin films of LDHs on iron substrates.

(1) The divalent cations provide necessary components to meet the basic structural requirement of LDHs, where the positive layers of LDHs usually consist of both divalent and trivalent cations. Trivalent cations (i.e., Fe^{3+}) can be generated *in situ* from iron plates in the corrosive solution. Detailed discussion has been involved in the manuscript (or see below).

"...We further attempted to create the corrosive solutions using other cations with lower or higher valence states, such as Na^+ , K^+ , Cr^{3+} , Ga^{3+} and Sn^{4+} , in lieu of divalent cations, but failed to generate LDHs-based nanoarray thin films on iron plates (see Fig. S5-6 in SI). This should be because those cations do not meet the basic structural requirement of LDHs, where the positive layers of LDHs usually consist of both divalent and trivalent cations.³³ Trivalent cations (i.e., Fe^{3+}) can be produced *in situ* from iron plates in the corrosive solution. These results further demonstrate that the presence of divalent cations is necessary for the formation of LDHs on iron plates, instead of the commonly formed iron rusts..."

(2) The divalent cations, especially Ni^{2+} , create a suitable, weak acidic solution that is crucial for the formation of abundant grain boundaries in the LDH nanosheets. Detailed discussion has been involved in the manuscript (or see below).

"...We also studied the effects of concentration of Ni^{2+} (as a representative divalent cation) on the morphology of the resulting electrodes. In particular, we prepared the corrosive solutions of Ni^{2+} with a wide concentration range from 10 to 200 $\mu\text{mol/L}$, and correspondingly we fabricated a series of electrode materials by the "corrosion engineering" method. Structural characterization results (Fig. S7-8) reveal that, LDHs-based nanoarray thin films can be generated on iron plates over the whole concentration range from 10 to 200 $\mu\text{mol/L}$, but a high concentration of Ni^{2+} (typically $\geq 100 \mu\text{mol/L}$) is crucial for the generation of abundant grain boundaries (or low crystallinity) in the LDH nanosheets. The reason behind this phenomenon might lie in the more acidic environment created in the solution containing higher Ni^{2+} concentration (e.g., pH 5.88 for Ni^{2+} concentration of 100 $\mu\text{mol/L}$, pH 6.28 for Ni^{2+} concentration of 10 $\mu\text{mol/L}$). In view of the growth preference of LDHs, the more acidic environment should suppress the crystallization of LDHs more, finally leading to the formation of grain boundaries-enriched LDH nanosheets..."

Comment 3: What is the role of the CO_2 and/or CO_3^{2-} in the fabrication process of catalyst?

Response 3: Carbonate (CO_3^{2-}) ions, which are generated from CO_2 *in situ*, are present as the intercalated anions for the formation of LDHs thin films in the fabrication process of catalyst.

We add the below sentences in the revised manuscript (please see page 7 in the revised manuscript).

“...This result is in agreement with the recent studies demonstrating the formation of carbonate-intercalated LDHs always preferred when CO_2 , which would generate carbonate ions *in situ*, was not intentionally avoided in the reaction system.^{31,32} ...”

Comment 4: For the preparation of the corrosive solution, a certain amount of divalent cations ($\text{NiSO}_4 \cdot 6\text{H}_2\text{O}$, $\text{Ni}(\text{NO}_3)_2 \cdot 6\text{H}_2\text{O}$, $\text{NiCl}_2 \cdot 6\text{H}_2\text{O}$, $\text{CoSO}_4 \cdot 7\text{H}_2\text{O}$, $\text{MnSO}_4 \cdot \text{H}_2\text{O}$ or MgSO_4 ; 10 mmol) was added into 100 mL deionized water to generate the corrosive solutions. The authors declare it is “a neutral aqueous solution”. Please check, without neutralization most of the solutions should be acidic, except the one of MgSO_4 .

Response 4: We thank the reviewer’s insightful comments. We measured all the aqueous solutions containing $\text{NiSO}_4 \cdot 6\text{H}_2\text{O}$, $\text{Ni}(\text{NO}_3)_2 \cdot 6\text{H}_2\text{O}$, $\text{NiCl}_2 \cdot 6\text{H}_2\text{O}$, $\text{CoSO}_4 \cdot 7\text{H}_2\text{O}$, $\text{MnSO}_4 \cdot \text{H}_2\text{O}$ or MgSO_4 (10 mmol). Their pH values are 5.88, 5.42, 6.11, 5.90, 5.35 and 6.83, respectively. These results show that the solution containing MgSO_4 is near neutral and other solutions are acidic, as predicted by the reviewer.

The pH values of these aqueous solutions have been involved in the revised manuscript (Methods 1.1). In addition, “a neutral aqueous solution” has been modified to “an aqueous solution” in the main text.

Comment 5: NiFe based material is a common material in the system under consideration, the authors should focus on defects and morphology introduced here. What is the actual formation mechanism of such a Fe-bearing structure?

Response 5: Thanks for the reviewer’s valuable question.

(1) In the revised manuscript, we provide an explanation on the oriented growth of nanosheet array thin films of LDHs (or see below).

“...The oriented growth of nanosheet array thin films of LDHs on iron plates can be explained by the classic heterogeneous nucleation/growth mechanism.³⁴ The iron corrosion can

吉林大学 化学学院

Department of Chemistry

Jilin University

Changchun 130012, China

Dr. Prof. Xiaoxin Zou

State Key Lab. Inorg. Synth. & Prep. Chem.

Tel: +86-431-85168221

xxzou@jlu.edu.cn

http://zouxxgroup.com/

produce a large amount of Fe^{3+} and OH^- ions (the precursors for LDHs nucleation) near the surface of iron plates and the heterogeneous nucleation of LDHs is kinetically favorable at the iron-solution interface. Due to the anisotropic crystal structure of LDHs, their growth in the *ab*-direction is faster than that in the *c*-direction,³⁴ so that all the LDH nanosheets tend to preferentially grow along the direction parallel to the basal planes of LDHs...”

(2) In the revised manuscript, we provide an explanation on the generation of abundant grain boundaries (or defects) in LDH nanosheets (or see below).

“...We also studied the effects of concentration of Ni^{2+} (as a representative divalent cation) on the morphology of the resulting electrodes. In particular, we prepared the corrosive solutions of Ni^{2+} with a wide concentration range from 10 to 200 $\mu\text{mol/L}$, and correspondingly we fabricated a series of electrode materials by the “corrosion engineering” method. Structural characterization results (Fig. S7-8) reveal that, LDHs-based nanoarray thin films can be generated on iron plates over the whole concentration range from 10 to 200 $\mu\text{mol/L}$, but a high concentration of Ni^{2+} (typically $\geq 100 \mu\text{mol/L}$) is crucial for the generation of abundant grain boundaries (or low crystallinity) in the LDH nanosheets. The reason behind this phenomenon might lie in the more acidic environment created in the solution containing higher Ni^{2+} concentration (e.g., pH 5.88 for Ni^{2+} concentration of 100 $\mu\text{mol/L}$, pH 6.28 for Ni^{2+} concentration of 10 $\mu\text{mol/L}$). In view of the growth preference of LDHs, the more acidic environment should suppress the crystallization of LDHs more, finally leading to the formation of grain boundaries-enriched LDH nanosheets...”

Comment 6: The atomic ratio plays a key role in determining the catalytic performance of NiFe based OER catalyst. The author should provide more information of Ni:Fe atomic ratio of O₂-Cat-2.

Response 6: We appreciate the reviewer’s insightful comment.

The Ni:Fe atomic ratio of O₂-Cat-2 is about 1.9:1. This Ni:Fe atomic ratio is similar with that reported in the original study (*Chem. Sci.* 2015, 6, 6624-6631). This result has been involved in the revised manuscript.

In addition, although O₂-Cat-1 (1.1:1 Ni:Fe for this material) and O₂-Cat-2 has different Ni:Fe atomic ratio, this difference in Ni:Fe atomic ratio might not account for the better catalytic activity of O₂-Cat-1 because the composition of both materials were in the optimal range for NiFe-based oxygen evolution electrocatalysts (*J. Am. Chem. Soc.* 2013, 135, 12329-12337, *J. Am. Chem. Soc.* 2015, 137, 1305-1313). A simple discussion has been added in the revised manuscript or see below.

“...The different Ni:Fe atomic ratio in two materials might not account for the better catalytic activity of O₂-Cat-1 because the composition of both materials were in the optimal range for NiFe-based oxygen evolution electrocatalysts.^{7,38}...”

吉林大学 化学学院

Department of Chemistry

Jilin University

Changchun 130012, China

Dr. Prof. Xiaoxin Zou

State Key Lab. Inorg. Synth. & Prep. Chem.

Tel: +86-431-85168221

xxzou@jlu.edu.cn

<http://zouxxgroup.com/>

Comment 7: Fig. 4a shows the polarization curves of O₂-Cat-1 and O₂-Cat-2 for OER. If O₂-Cat-1 and O₂-Cat-2 show similar LDH structure, contain the same Ni:Fe atomic ratio, the intensity of the redox peaks of Ni(II)/Ni(III or IV) should be similar to each other. Why they are totally different?

Response 7: Thanks. Before our response to the reviewer's question, we want to show two facts first for the reviewer.

(1) Different from O₂-Cat-1, O₂-Cat-2 does not exhibit obvious the pre-oxidation peak of Ni. Our result is in agreement with that reported in the original study (*Chem. Sci.* 6, 6624-6631).

(2) O₂-Cat-1 and O₂-Cat-2 have similar LDH structure, as revealed by some structural characterizations, such as SEM and TEM results (detailed discussion can also be found in original study *Chem. Sci.* 6, 6624-6631). But O₂-Cat-1 and O₂-Cat-2 contain different Ni:Fe atomic ratios. They contain Ni:Fe atomic ratios of 1.1:1 and 1.9:1, respectively.

However, different Ni:Fe atomic ratio in two materials seems not explain why O₂-Cat-2 does not exhibit obvious the pre-oxidation peak of Ni. This is because Ni₃Fe-based water oxidation catalysts with a wide range of Ni:Fe atomic ratio usually show the pre-oxidation peak of Ni before the oxygen evolution reaction (see references: *e.g.*, *J. Am. Chem. Soc.* 2013, 135, 12329; *J. Am. Chem. Soc.* 2017, 139, 11361). We speculate that this phenomenon should correlate with the surface microenvironment of Ni species in the material, which ultimately originates from the particular electrochemical synthetic method. But the clear reasons (or atomic basis) for this phenomenon are still unclear at current stage.

Comment 8: In the introduction, the authors conclude that the deactivation of an OER catalyst during OER can be caused by various adverse microstructural evolutions of catalytic active phases, such as oxidative decomposition, structural reconstruction, metal leaching, irregular aggregation and the peeling of catalytic active species from the current collector, especially at large catalytic current densities. The author should analyze the stability of the catalyst from the five factors during the long-time electrolysis process.

Response 8: We appreciate the reviewer's good suggestion. We have provide some additional discussion on the stability of the catalyst in the revised manuscript (or see below).

"...The above results overall demonstrate the meliority of our "corrosion engineering" method over the electrochemical deposition method and other methods reported recently (Table S1 in SI) in terms of making highly efficient oxygen evolution electrodes. O₂-Cat-1's excellent electrocatalytic performance can be attributed to the following several facts. On the one hand, O₂-Cat-1 has the common structural advantages of binder-free nanoarray electrodes and LDHs

吉林大学 化学学院

Department of Chemistry

Jilin University

Changchun 130012, China

Dr. Prof. Xiaoxin Zou

State Key Lab. Inorg. Synth. & Prep. Chem.

Tel: +86-431-85168221

xxzou@jlu.edu.cn

<http://zouxxgroup.com/>

materials. (i) The well-oriented nanosheet array architecture is beneficial for avoiding the irregular aggregation of catalytic active phase, exposing the catalytic active sites, and facilitating the mass transport during OER (*i.e.*, allowing better supply of reactants to get the active sites and evacuation of gaseous products from the reaction system; see Fig. S15 and a movie in SI). (ii) The binder-free structural feature (*i.e.*, binding its catalytically active LDH thin film to the underlying iron substrate without a polymer binder) leads to the presence of an intimate contact between the catalytically active LDH thin film and the current collector (*i.e.*, the underlying iron substrate), further eliminating the interfacial resistance between the two to a large extent. (iii) LDHs material usually have good structural stability against oxidative decomposition, structural reconstruction and metal leaching in alkaline media. On the other hand, O₂-Cat-1 possesses two particular structural advantages that are not achieved by the electrodes from other synthetic methods (*e.g.*, electrochemical deposition). (i) O₂-Cat-1 comprises the LDH nanosheets with ultrasmall crystalline domains and abundant grain boundaries, giving rise to a large proportion of highly active catalytic sites. (ii) O₂-Cat-1 has the catalytically active LDH thin film that strongly adheres to the iron substrate, avoiding the peeling of catalytically active phases from the current collector during long-term electrocatalysis, even at large current densities (also see below).”

Reviewer: 3

Comment 1: In this work, Zou et al report the spontaneous corrosion reaction of iron to fabricate efficient oxygen evolution electrode without needing any additional energy inputs (heat or electricity). The resulting electrode is among the most active electrodes for OER in alkaline electrolyte reported to date. Moreover, the resulting electrode exhibits a record long-term catalytic stability for more than 6000 hours, even at a large current density of 1000 mA/cm². Both the synthetic method and the record long-term catalytic stability in this work are impressive. Thus, I recommend the publication of this paper in Nature Communications after some revisions.

Response 1: We appreciate the reviewer’s positive comments about our work.

Comment 2: How the authors got rid of the bubbles effect. At high current densities, the formation of bubbles usually generates very noisy LSV data but the curves shown in this paper are very smooth and precise.

Response 2: We did not use any special method to get rid of the bubbles and overcome the so-called bubble effect. We found that a large volume of oxygen gas was generated on the electrode, and the oxygen gas could quickly evacuate from the electrode even at high current densities (see the video we have included in SI). This should be the reason why the LSV curve of

OER appears smooth. Needless to say, this also indicates that the electrode might have conducive surfaces that are beneficial for the evacuation of the oxygen gas forming by the OER.

Comment 3: It should be better to give the mass loadings of catalysts (*i.e.*, LDH species).

Response 3: The mass loadings of LDH species are provided in the revised manuscript (see Methods 1.1).

Comment 4: As described in the paper, when iron substrates are brought in contact with water and air, iron is ready to be corroded. Could the authors give some explanation why the electrodes (Cat-O₂-1 and 3D-Cat-O₂-1) could be so stable during OER? In other words, could the underlying iron substrates in these electrodes be further corroded by oxygen during OER?

Response 4: Thanks for the reviewer's valuable question! The underlying iron substrates in the electrodes do not further corroded by oxygen during OER (see structural characterizations on the electrode after OER testing, Figure S14 in SI). This is because the underlying iron substrates are covered with stable thin films of LDHs, protecting the iron substrates against further corrosion.

We add the below sentences in the revised manuscript.

“These results also demonstrate that the underlying iron substrates do not further corroded by oxygen during OER. This should be because the underlying iron substrates are covered with stable thin films of LDHs, protecting the iron substrates against further corrosion.”

Comment 5: Please provide the characterization results (*e.g.*, SEM images) of 3D-Cat-O₂-2.

Response 5: The digital and SEM images of 3D-Cat-O₂-2 are provided in Supporting Information (see Figure S17, SI).

Comment 6: In addition, IrO₂ is widely considered as the benchmarking OER electrocatalyst because of its high catalytic activity in acidic and alkaline media. However, this catalyst is not stable enough in alkaline electrolyte. This is one of the reasons why IrO₂ is not used in alkaline electrolyzer. The authors should point out this drawback of IrO₂ in the paper to show more complete information for the readers.

Response 6: We appreciate the reviewer's insightful comments here! We add the below sentence in the revised manuscript (see Methods 1.2).

“Although IrO₂ is widely used as the benchmark electrocatalyst for OER, this material is not catalytically stable enough in alkaline media. Thus, the long-time stability of IrO₂ for OER was not measured here.”

吉林大学 化学学院

Department of Chemistry

Jilin University

Changchun 130012, China

Dr. Prof. Xiaoxin Zou

State Key Lab. Inorg. Synth. & Prep. Chem.

Tel: +86-431-85168221

xxzou@jlu.edu.cn

<http://zouxxgroup.com/>

We hope that we have addressed the comments and suggestions raised by the reviewers. We look forward to hearing your decision at your convenience please. Thanks a lot!

With best regards,

Sincerely yours,

Xiaoxin Zou

Associate Professor of Chemistry

State Key Laboratory of Inorganic Synthesis and Preparative Chemistry, College of Chemistry,
Jilin University, Changchun 130012, P. R. China

Reviewers' comments:

Reviewer #1 (Remarks to the Author):

The authors have addressed the technical questions well, however my overall impression about the novelty of this paper has not changed much. The authors made arguments about the novelty against previous NiFe catalysts, in terms of long term stability (>6000h), and high current density (~1000 mA cm⁻²). The stability of anode is not a very big issue in industrial alkaline electrolyzers. The control experiments shown in the paper is not convincing, many more studies have shown NiFe are stable. To reach current density ~1000 mA cm⁻² is not practical. Rarely any industrial alkaline electrolyzers can run at that current density, even the half would be great.

The corrosion engineering method is not very new, this reviewer has reviewed several papers recently on this topic, for example, a recent paper using the corrosion of Ni foam to fabricate NiFe catalysts (Chem. Commun., 2018,54, 3262-3265).

Overall, as I mentioned in the first report, this is a good paper, but slightly below my bar for Nat Commun.

Reviewer #2 (Remarks to the Author):

The authors have addressed most of the issues raised in the first round review. I recommend the publication of this paper in Nature Communications after addressing the following additional points.

- 1, Please synthesize a catalyst without the presence of CO₂ and/or CO₃²⁻ for comparison.
- 2, During long-time electrolysis process at high current density, metal leaching is a big issue which leads to the deactivation of an OER catalyst. Please show the concentration of Fe and Ni ions in the electrolyte, which may change over time during the electrolysis process.

Reviewer #3 (Remarks to the Author):

This work reports the spontaneous corrosion reaction of iron to fabricate efficient oxygen evolution electrode materials. The resulting electrode is among the most active electrodes for OER in alkaline electrolyte reported to date. The impressive long-term catalytic stability is promising for the future applications in water splitting. The revised manuscript has been well prepared based on the comments of reviewers. Thus, I recommend the publication of this paper in Nature Communications without any further revisions.

Reviewer's Comments and Our Responses:

Reviewer: 1

Comment 1: The authors have addressed the technical questions well, however my overall impression about the novelty of this paper has not changed much. The authors made arguments about the novelty against previous NiFe catalysts, in terms of long term stability (>6000 h), and high current density ($\sim 1000 \text{ mA cm}^{-2}$). The stability of anode is not a very big issue in industrial alkaline electrolyzers. The control experiments shown in the paper is not convincing, many more studies have shown NiFe are stable. To reach current density $\sim 1000 \text{ mA cm}^{-2}$ is not practical. Rarely any industrial alkaline electrolyzers can run at that current density, even the half would be great.

The corrosion engineering method is not very new, this reviewer has reviewed several papers recently on this topic, for example, a recent paper using the corrosion of Ni foam to fabricate NiFe catalysts (Chem. Commun., 2018, 54, 3262-3265).

Overall, as I mentioned in the first report, this is a good paper, but slightly below my bar for *Nat Commun*.

Response 1: We thank this reviewer for his/her comments here! We also appreciate the reviewer for bringing this important paper about NiFe catalysts to our attention. We have cited this paper in the manuscript (please see reference 10).

We read this related paper carefully, and we found that:

(1) In that paper, iron-doped nickel hydroxide supported on nickel foam was reported. Thus, the material in that paper is different from our materials in this work (*i.e.*, NiFe-LDHs grown on iron substrates).

(2) In that paper, the material is not quite stable. As shown in Figure 4c, the material underwent detectable, continuous activation loss over 100-hours-long electrocatalysis at 200 mA/cm^2 . Thus, we would like to claim again that developing highly active oxygen evolution electrodes that can possess significantly prolonged catalytic lifetime (*e.g.*, beyond thousands of hours) still remains a great challenge.

(3) In that paper, the surficial etching of nickel foam by HCl is crucial for the formation of iron-doped nickel hydroxide on nickel foam. The so-called “corrosion engineering” method in that paper is totally different from the method presented in our work (*i.e.*, totally different corrosion mechanism between them).

Reviewer: 2

Comment 1: The authors have addressed most of the issues raised in the first round review. I recommend the publication of this paper in Nature Communications after addressing the following additional points.

Response 1: We thank the reviewer's additional comments here! We have addressed the comments point-by-point as follows.

Comment 2: Please synthesize a catalyst without the presence of CO_2 and/or CO_3^{2-} for comparison.

Response 2: As suggested by the reviewer, we attempted to synthesize the material in the absence of CO_2 and/or CO_3^{2-} . In particular, we synthesized a material under a pure oxygen atmosphere. The results reveal that the presence of CO_2 and/or CO_3^{2-} is not necessary for the formation of LDHs on the iron substrates (see Supplementary Figure 10). However, recent studies show that when measured in the alkaline aqueous electrolyte in ambient air, the active catalysts for the OER are always the carbonate-containing LDHs, regardless of the precatalyst composition (*Energy Environ. Sci.* 2016, 9, 1734).

Comment 3: During long-time electrolysis process at high current density, metal leaching is a big issue which leads to the deactivation of an OER catalyst. Please show the concentration of Fe and Ni ions in the electrolyte, which may change over time during the electrolysis process.

Response 3: Thanks! In order to determine whether Fe and Ni ions were leached in the electrolyte at high current density, we used ICP-OES to detect the Fe and Ni ions in the electrolyte after the long-time electrolysis process at 1000 mA/cm^2 . The result shows that no detectable leached metallic species are present in the electrolyte (see Page S4 and S17 in Supplementary Information, Supplementary Table 2).

Reviewer: 3

Comment 1: This work reports the spontaneous corrosion reaction of iron to fabricate efficient oxygen evolution electrode materials. The resulting electrode is among the most active electrodes for OER in alkaline electrolyte reported to date. The impressive long-term catalytic stability is promising for the future applications in water splitting. The revised manuscript has been well

吉林大学 化学学院

Department of Chemistry

Jilin University

Changchun 130012, China

Dr. Prof. Xiaoxin Zou

State Key Lab. Inorg. Synth. & Prep. Chem.

Tel: +86-431-85168221

xxzou@jlu.edu.cn

<http://zouxxgroup.com/>

prepared based on the comments of reviewers. Thus, I recommend the publication of this paper in Nature Communications without any further revisions.

Response 1: We appreciate the reviewer for recommending the publication of our work. Many thanks!

We hope that we have addressed the comments and suggestions raised by the reviewers. We look forward to hearing your decision at your convenience please. Thanks a lot!

With best regards,

Sincerely yours,

Xiaoxin Zou

Associate Professor of Chemistry

State Key Laboratory of Inorganic Synthesis and Preparative Chemistry, College of Chemistry, Jilin University, Changchun 130012, P. R. China

REVIEWERS' COMMENTS:

Reviewer #2 (Remarks to the Author):

The authors addressed my concerns adequately, and I recommend this manuscript for publication in Nat.Comm.

吉林大學 化學學院

Department of Chemistry

Jilin University

Changchun 130012, China

Dr. Prof. Xiaoxin Zou

State Key Lab. Inorg. Synth. & Prep. Chem.

Tel: +86-431-85168221

xxzou@jlu.edu.cn

<http://zouxxgroup.com/>

Our Responses to the referees' comments:

Reviewer: 2

Comment 1: The authors addressed my concerns adequately, and I recommend this manuscript for publication in Nat.Comm.

Response 1: We thank the reviewer for recommending the publication of our paper!